# Evaluation of Modified Atmosphere Packaging in Combination with Active Packaging to Increase Shelf Life of High-in Beta-Glucan Gluten Free Cake

**DOI:** 10.3390/foods11060872

**Published:** 2022-03-18

**Authors:** Jarosław Wyrwisz, Sabina Karp, Marcin Andrzej Kurek, Małgorzata Moczkowska-Wyrwisz

**Affiliations:** 1Institute of Human Nutrition Sciences, Department of Technique and Food Product Development, Warsaw University of Life Sciences, 159c Nowoursynowska St., 02-776 Warsaw, Poland; sabina_karp@sggw.edu.pl (S.K.); marcin_kurek@sggw.edu.pl (M.A.K.); 2Institute of Human Nutrition Sciences, Department of Food Gastronomy and Food Hygiene, Warsaw University of Life Sciences, 159c Nowoursynowska St., 02-776 Warsaw, Poland; malgorzata_moczkowska@sggw.edu.pl

**Keywords:** active packaging, essential oils, water activity, microbiological quality, texture, color

## Abstract

Modified atmosphere packaging and active packaging were combined to prolong the shelf life and quality of the clean label, gluten-free (GF), yeast-leavened cakes enriched in oat fiber preparation. Star anise, cinnamon bark, and clove essential oils were used as emitters of active substances. The following concentrations of gases were chosen: 0% CO_2_/100%/N_2_ (MAP_1_), 60% CO_2_/40% N_2_ (MAP_2_), and approx. 78% N_2_/21% O_2_/0.04% CO_2_ (ATM). Microbiological and physicochemical analyses were conducted. GF cakes were stored for 14 days (analysis in 0, 7, and 14 days). The results showed a decrease in moisture content and lightness of crumb and an increase in hardness. EOs significantly (*p* ≤ 0.05) slowed down the growth of microorganisms regardless of the type of gas mixture. However, the best bacteriostatic effect was in MAP_2_. The content of beta-glucan did not change throughout the storage time. Generally, the best results were obtained with the combination of MAP and active packaging—60% of CO_2_ and 40% of N_2_—where cinnamon or clove essential oils were used.

## 1. Introduction

Gluten-free (GF) products are in great demand in the food industry of Western countries due to the increasing number of people with celiac disease and other gluten-related illnesses. GF products are still characterized by poor structure, volume, little acceptable taste, and inefficient dietary fiber, vitamin, and mineral content [1]. Changes in bakery and pastry recipes can improve their crumb’s structure. The addition of hydrocolloids or dietary fibers can be a primary solution. An example of dietary fiber with structure-forming properties is soluble oat beta-glucan. In aqueous solutions, it creates a gel and increases viscosity. Some studies about the application of oat beta-glucan into bakery production aimed at different beta-glucan pretreatments before incorporating it into the recipe [2]. The application of beta-glucan was possible in products based on wheat flour and in gluten-free production [3]. Moreover, oat beta-glucan has many health-promoting properties [4].

Another important issue regarding gluten-free products is maintaining their freshness and prolonging shelf life. The shelf life of bakery goods is limited by fungal growth, staling, and moisture loss. There is also a risk of recontamination during cooling and packaging [5]. The main microbes that spoil bakery products *are Aspergillus* sp., *Penicillium* sp., and *Rhizopusstolonifer*. Those fungi are responsible for off-flavors, mycotoxins, and bad appearance [6]. Different approaches can protect food from fungal and spoilage growth and minimalize storage degradation lipid oxidation [7]. Consumers’ demand for fresh, tasty, convenient food with prolonged shelf-life is the main impulsion for developing new food packaging technologies [8,9]. For now, modified atmosphere packaging (MAP)—intelligence or active packaging—is one of the most popular solutions [10,11]. The new active packaging approach uses emitters of active substances to prolong the shelf life of products by limiting microbial growth and decreasing oxidation processes. It is possible to use essential oils (EOs) derived from plants as an active agent of natural origin [12]. They are known for antimicrobial activity and, additionally, are registered as GRAS (Generally Recognized as Safe) by the United State Food and Drug Administration. EOs are natural alternatives to synthetic fungicides that can be incorporated within the package walls or in separate sachets [13,14]. Generally, EOs are composed of various components such as phenols, aldehydes, ketones, terpenes, carbohydrates, ethers, and alcohols responsible for outstanding biological activities [15]. EOs also have antioxidant properties, making them a promising natural food preservative [16]. Antioxidant properties are responsible for most phenolic compounds that interact with the food matrix and, as a result, protect food from microbial growth [17,18]. Several studies showed their effectiveness in microbial inhibition. Garlic (*Allium sativum*) essence oil was incorporated into plastic films and tested against *Listeria monocytogenes*, *Escherichia coli*, and *Brochothrix thermosphacta* in beef [19]. Cumin (*Cuminum cyminum* (L.)) EO was used to prolong the shelf life of stored wheat and chickpeas [7]. Cinnamon and clove EOs were examined for their inhibitory activity against different spoilage microbials of intermediate-moisture foods under modified atmosphere conditions [20]. Star anise (*Illicium verum*) and black caraway (*Carum nigrum*) EOs were also reported as strong natural antioxidants [21]. EOs show both qualitative and quantitative variations due to a great diversity of bioactive compounds resulting in variable biological effectiveness. The strong aroma is another hindrance in using EOs in food, limiting their application. However, the proper aroma of EOs can be controlled via selecting EO carefully according to the kind of food in question. In addition to direct contact, EOs can exhibit their function through active food packaging [22].

In the literature, there are a lot of articles regarding MAP and active packaging. Nevertheless, there is not enough to focus on combining these two types of packaging methods in GF bakery. Moreover, there are already food products, mainly meat products, packed in MAP on the market. Active and intelligent packaging also started to appear. This fact indicates that different packaging systems are highly needed in the food industry. The combination of EOs with modified atmosphere packaging may cause the synergistic effect of these factors because, as many publications indicate [5,20], the individual influence of these factors is not satisfactory to inhibit the growth of microorganisms. Therefore, the study aimed to evaluate the combined packaging system—MAP (modified atmosphere packaging) and active packaging GF leavened cakes enriched in high-in beta-glucan oat fiber powder. Three types of essential oils were used as emitters of active substances: star anise (*Illicium verum*) oil, cinnamon bark (*Cinnamomum zeylanicum*) oil, and clove (*Eugenia caryophyllata*) oil. These EOs were chosen because of their documented properties and common availability and usage in food preparation. The use of MAP in combination with EOs will prolong the shelf life and the quality of clean label GF cakes.

## 2. Materials and Methods

### 2.1. Materials

A local market supplied the ingredients for batter formulation—rice flour (on 100 g of flour: 6 g proteins, 78 g carbohydrates, 1 g fat, 1.4 g fiber, Melvit S.A, Warsaw, Poland), corn flour (on 100 g of flour: 9 g proteins, 70 g carbohydrates, 4 g fat, 6 g fiber, Melvit S.A, Warsaw, Poland), corn starch (Bezgluten, Koniusza near Krakow, Poland), tapioca starch (Thailand), instant dried yeasts (Lesaffre, Marcq-en-Baroeul, France), rapeseed oil (Olej Kujawski, Kruszwica, Poland), sugar (Diamant, Poznań, Poland), salt (o’ Sole, Września, Poland), eggs, and water. Other materials were high-in-β-glucan oat fiber powder (44 g/100 g of dietary fiber—23 g soluble and 21 g insoluble fraction; 16 g/100 g of β-glucan; 16 g proteins, 23 g carbohydrates, 7 g fat) (Microstructure, Warsaw, Poland). Three types of 100% pure essential oils (steam distilled) (EOs) were purchased from NOW (Bloomingdale, IL, USA)—star anise (*Illicium verum*) oil (AE), cinnamon bark (*Cinnamomum zeylanicum*) oil (CN), and clove (*Eugenia caryophyllata*) oil (CV).

### 2.2. Batter Formulation and Cake Baking

Based on a preliminary study referred to optimization of batter formulation, five sets of gluten-free (GF) yeast-leavened cake with 13.62% of oat dietary fiber powder were prepared in the following manner. Values in parentheses show percentage amounts of each ingredient in the formulation *w*/*w* to flour–starch mix. Yeast (1.82%), sugar (34.06%), and water (61.49%) were gently stirred and waited for 2 min. Then, dry ingredients—rice flour (41.66%), corn flour (24.98%), corn starch (16.68%), tapioca starch (16.68%), oat dietary fiber powder (13.62%), and eggs (24.98%)—were added. All were mixed for 2 min. Then, salt (0.34%) and oil (17.03%) were gradually added and continuously mixed for another 5 min. After that, the batter was proofed for 10 min, then divided into equal portions (250 g per baking mold) and put to proofing conditions (37 °C and 80% relative humidity) again for 30 min. In the end, the batter was baked for 40 min at 160 °C in a convection oven (CPE 110, Küppersbuch, Germany) and cooled to ambient temperature. Two hours after baking, one set of cakes was analyzed (D0), and the other four were packed.

### 2.3. Packaging

After cooling down, freshly baked GF cakes were packed in a combined system of modified atmosphere packaging (MAP) and active packaging (AP), with the usage of essential oils (EOs)—star anise oil (AE), cinnamon bark oil (CN), and clove oil (CV). At first, PET/PE trays (187 mm × 137 mm × 83 mm) with an attached rectangle of filter papers (60 mm × 20 mm) were sterilized under a UV lamp (power 36 W) for 1 h. Then, the cakes were placed into trays. After EOs were put onto filter paper (500 µL of concentrated EO), they were packed with an M3 packaging machine (Sealpack, Oldenburg, Germany) using PET/CPP + AF film 44 μm thick with maximum oxygen permeability not exceeding 10 cm^3^/m^2^/24 h/bar at 23 °C with a relative humidity of 0% (EC04, Corenso, Helsinki, Finland). Mixtures of carbon dioxide (CO_2_), oxygen (O_2_), and nitrogen (N_2_) gases were prepared in the following concentrations: 0% CO_2_/100%/N_2_ (MAP_1_), 60% CO_2_/40% N_2_ (MAP_2_), and approx. 78% N_2_/21% O_2_/0.04% CO_2_ (ATM). The gas mixtures were obtained in a gas mixer (KM 30.3 ME, Witt-Gasetechnik, Germany). As a reference, one sample set was not packed with EOs (NO). Samples were stored for 7 (D7) and 14 (D14) days at ambient temperature (20 ± 2 °C).

### 2.4. Microbiology

The microbiology analysis was carried out to examine the ability of used EOs to inhibit the growth of pathogens. Microbiological analysis of the GF cakes was carried out every 7 days of storage, including total plate count (TPC). GF cake samples were taken from each packaging system and transported to an accredited laboratory, where measurements were performed in triplicate (*n* = 3) following the PN-EN ISO 4833-1:2013-12 standard for TPC. The determination of the number of microorganisms capable of growth and colony formation in the solid medium after incubation under aerobic conditions at 30 °C was performed by a horizontal method. Results were expressed as log10 cfu/g GF cake.

### 2.5. Moisture Content and Water Activity

Both crumb and crust of GF cake in the form of small particles (to facilitate water evaporation) were taken to measure moisture content and water activity. Moisture content was determined with moisture analyzer MA 60.3Y (Radwag, Radom, Poland), and the water activity was measured by the dew point methodology using Water Activity Meter 4 TE (Aqualab, Pullman, WA, USA). Both measurements were performed in triplicate (*n* = 3).

### 2.6. Beta-Glucan Content

The beta-glucan content was measured with β-glucan Assay Kit (Mixed Linkage), following the manufacturer’s instructions (Megazyme, Bray, Ireland). Spectrophotometer Shimadzu UV-1800 (Shimadzu Inc., Kyoto, Japan) was used to measure absorbance at 510 nm. Measurements were performed in triplicate (*n* = 3).

### 2.7. Texture

Physical properties were assessed to describe the quality of the baked gluten-free cake based on previous studies [3,23]. Texture profile analysis (TPA) was determined using a universal testing machine (Instron 5965; Instron, Canton, MA, USA) with Bluehill 2 software installed. Samples were cut into 2 cm × 2 cm × 2 cm cubes, and double compression tests were performed in eight repetitions (*n* = 8) for each sample. A 50 mm cylindrical probe, 50% compression of sample, and 5 s relaxation time were used. A 50 N load cell was used, and the crosshead speed was set at 120 mm/min.

### 2.8. Color

The color was examined in CIE L*a*b* measuring system with Minolta CR-400 colorimeter (Konica Minolta Inc., Osaka, Japan) using an 8 mm aperture, illuminate D_65_ (6500 K color temperature) at a standard observation of 2°. Colorimeter was calibrated against a white plate (L* = 98.45, a* = −0.10, b* = −0.13). L* (lightness ranged from 0 to 100), a* (color axis ranged from greenness (−a*) to redness (+a*)), and b* (color axis ranged from blueness (−b*) to yellowness (+b*)). Samples were cut into slices, and then color was measured 10 times (*n* = 10) in different spots.

### 2.9. Statistical Analysis

The effect of different types of EOs (NO, AE, CV, CN), gas mixtures (ATM, MAP_1_, MAP_2_), and storage times were analyzed. Verification of difference significance of investigated parameters was studied using ANOVA with applied Fisher test with a significance level of α = 0.05. The software was provided by Statistica version 13 (StatSoft, Inc., Tulsa, OK, USA).

## 3. Results and Discussion

### 3.1. Microbiology

To extend shelf life and freshness, which are dependent on microbial growth, it is very essential to adjust the proper packaging method to the specific foods. In this study, the main principle was the combined usage of MAP and active packaging method (with EOs) to pack gluten-free cakes. Gluten-free products, especially those with a high fiber content, are very difficult to store for a long time. The analysis of total plate count was conducted, and the results are presented in Table 1. The limiting result of total plate count was 5.0 log (cfu/g), which indicates that those samples were unfit for human consumption. The best results (the lowest TPC) were obtained in samples with storage of 7 d, packed in MAP_2_ for CN (0.54 log [cfu/g]; *p* ≤ 0.05), AE, and CV (0.78 log [cfu/g] and 0.83 log [cfu/g]; *p* > 0.05; respectively). The worst type of packaging was ATM regardless of the type of EO (TPC > 2.3 log [cfu/g] and TPC > 3.2 log [cfu/g] for samples stored 7 and 14 d, respectively). The reference samples packed without EOs (NO) had the highest total plate count regardless of the type of gas mixture used. In another study, the antimicrobial effects of cinnamon EOs on the shelf-life of kolompe (Iranian cookie) were compared to the yeast, mold, total count, *E. coli*, *Enterobacteriaceae*,—positive *Staphylococacci coagulase,* and *Bacillus cereus* of samples containing different concentrations essential oil with the control samples. Aerobic microorganisms, yeast, or mold did not grow on the samples even on the 30th day [24]. This experiment proved that the best-performing results are for the combination of MAP and active packaging. Anaerobic conditions caused a significant reduction (*p* ≤ 0.05) in microbial growth in both MAP_1_ and MAP_2_, and the presence of EOs enhanced this effect. The best gas composition for GF yeast-leavened cake—stored for 7 and 14 d—was 60% CO_2_ and 40% N_2_, where CN (TPC_D7_ = 0.54 log [cfu/g] and TPC_D14_ = 0.84 log [cfu/g]; *p* ≤ 0.05) or CV (TPC_D7_ = 0.83 log [cfu/g] and TPC_D14_ = 0.94 log [cfu/g]; *p* > 0.05) essential oils were used. CO_2_ had bacteriostatic activity, while aerobic microorganisms and mold are also sensitive to a high-CO_2_ atmosphere [25]. The differences between the inhibiting activity of EOs can be associated with the concentration of those EOs that show the best power. Each EO may be the most active at different concentrations. Those findings are in line with Gutiérrez [5], who experimented on gluten-free bread loaves, which were packed in MAP in combination with active packaging. Using CN essential oil obtained the best results for samples packed with the combined method, and the application of only MAP or only active packaging did not give sufficient microbial growth inhibition. The release of essential oils to the package headspace caused absorption of the essential oils on the food surface, which delayed the growth of microorganisms [26]. Moreover, the thin layer of absorbed essential oils covering the food surface possibly interacted and reduced oxygen diffusion and delayed the growth of microorganisms [27]. Another study presented by Janjarasskul [28] focused on sponge cake packed in an active packaging system, where O_2_ scavenger and an ethanol emitter as alternatives to chemical preservatives were used. Again, the best results of delaying microbial growth were obtained in a combined system of packaging. So, the synergistic effect of both methods of active packaging had the strongest impact on microbiological quality. Prolonging of wheat bread shelf-life was also obtained by Muizniece-Brasava [29], who checked the combination of MAP (60% of CO_2_, 40% of N_2_), different film pouches, and iron-based oxygen scavenger sachets. The 28-day storage time was tested. The results showed that active packaging with oxygen emitter prevented microbial growth and extended shelf life for 14 days. If the quality of wheat bread is of concern, the best combination was Multibarrier 60 film, MAP (60% of CO_2_, 40% of N_2_), and oxygen scavenger. Ibrahium [30] tested the antimicrobial activity of clove essential oil on the preservation of cake by determining the counts of yeast, mold, coliforms, and total bacteria for 28 days. All of the tested microorganisms had less growth than control that decreased with the increasing levels of clove essential oil.

### 3.2. Moisture Content and Water Activity

Water activity and moisture content are two primary critical parameters regarding microbiological quality. Food products should have low water activity to slow down or inhibit microbial growth. Water activity below 0.8 is the safest [31]. The results are shown in Table 2. It can be seen that water activity ranged between 0.9324 and 0.9479 in D7, where differences were significant (*p* ≤ 0.05). In D14, it was between 0.9236 and 0.9432, where the significance varied, and the water activity tended to decrease. A similar trend was observed in moisture content, where the decrease was significant (*p* ≤ 0.05). The biggest change was observed in samples packed in ATM independently from EOs (*p* ≤ 0.05). This result proves that MAP prolongs the freshness of food. No significant change was observed for sample CV/MAP_2_ and CN/MAP_2_, indicating good moisture retention. A higher tendency to retain moisture during storage in a package with a higher CO_2_ content (in MAP_2_) than ATM and MAP_1_ was observed. However, no more significant microbial growth was observed. The presence of CO_2_ could have a greater bacteriostatic effect than the lower moisture content. This may be because CO_2_ dissociates on the surface to a greater extent with higher moisture content in the product, leading to the formation of carbonic acid on the product surface and a reduced pH value. Starch retrogradation and crystallization occur during storage time, which accelerates water migration out of the product. It has an impact not only on the texture of the baked product but also on water content during storage. Sánchez [32], in a study about the redistribution of water in gluten-free bread, also observed the decrease in crumb moister content after 7 days of storage. They also observed that water migrated from crumb to crust. Moreover, the process of recrystallization occurred. Water content and its redistribution influenced the kind of starch crystallites formed and the hardness of stored bread.

### 3.3. Beta-Glucan Content

Oat dietary fiber powder with high beta-glucan content was used not only as a structure-making agent but also to improve the nutritional value of the gluten-free cake. One of the goals of this study was to evaluate if the storage time influences the content of OBG. The results are presented in Table 2. The values of OBG stayed statistically unchanged (no significant differences, (*p* > 0.05)). This result is very important and proves that the pro-health benefits will be maintained during storage in the combined (MAP and active packaging) packaging system. There are not enough articles regarding the stability of beta-glucan during storage time that focus on the exact content of this compound throughout the storage time. However, the study conducted by Niba [33] refers to the beta-glucan and starch content during storage time (7 days) in cornbread in different conditions—ambient temperature, freezing, and refrigerating conditions. The results were very surprising because an increase in beta-glucan content was observed. It was explained by increased extractability and solubility because of structural changes in bread with storage, regardless of storage temperature.

### 3.4. Cake Quality Parameters

The color and texture of food products, especially baked goods, are highly important for consumers because they indicate good quality. Therefore, the components of those two parameters should be under special supervision. The results of color coordinates L*, a*, and b* are shown in Table 3. Where the lightness L* is concerned, a decrease in this color coordinate during storage time (*p* ≤ 0.05) was observed. In D7, samples packed in MAP_2_ with different EOs were significantly darker *p* ≤ 0.05) than others, while samples packed in ATM were substantially lighter (*p* ≤ 0.05). The darkest samples were those that were kept for 14 days in MAP_1_ and MAP_2_ with no EOs. There is no specific trend in saturation of redness and yellowness between samples. However, it can be observed that samples kept in ATM had lower saturation of redness on the 7th and 14th days. Similar findings were reported by Botosoa [34]. In their study, they monitored changes in sponge cake during 20-day storage at 20 °C. Samples of cake were packed in polyamide/polyethylene gas using a vacuum sealer. No significant changes in color were observed. Nevertheless, a slight decrease in lightness (no statistical significance) was noted, as well as an increase in yellow color. The authors suggested that in addition to the Maillard reactions that develop color and flavor during baking, another process affects the final product’s color. Lipid oxidation is meant to induce changes in the crumb color of the cake during the aging process. They also suggest that the observation time was too short because the above processes did not occur. Even more, the increased yellowness saturation related to fat and yolk globules became more concentrated upon dehydration that occurred during aging.

Hardness and springiness were considered to observe changes in texture during storage (Table 4). The addition of oat beta-glucan resulted in the crumb’s structure development. This was possible because oat beta-glucan can form a soft gel when bonded with water [35]. The packaging and storage conditions significantly affect the hardness of bakery and starch-based products. The difference in relative vapor pressure between food and the headspace is the primary driving force of moisture transfer and loss of moisture, which induced increasing hardness [36,37]. The biggest impact on hardness and springiness was seen from the packaging method and the composition of gases. The addition of EOs had a less significant effect. In general, the increase in hardness was observed throughout the storage time. The most rapid change was seen between D0 and D7 (*p* ≤ 0.05). The softest crumb of the cake had a sample in D0 (7.77 N; *p* ≤ 0.05). Samples AE/MAP_1_, CV/MAP_1_, and CN/MAP_1_ maintained values of texture between D7 and D14 the most, while samples AE/ATM, CN/ATM, and NO/ATM had the most rapid increase in hardness (from 15.65 N to 29.03 N; from 19.23 N to 36.61 N; from 19.75 N to 28.01 N; *p* ≤ 0.05; respectively). A similar trend was observed in the results of springiness. There was a decrease in this parameter in samples after 14 days of storage. In addition, samples CV/MAP_1_ and CN/MAP_1_ maintained springiness the most after 7 days of storage. Texture changes relate to the previously mentioned water migration and starch retrogradation. Despite high barrier packaging that prevents water loss, other mechanisms occurred simultaneously, so there is no explanation of texture changes. Janjarasskul [28] also observed the increase in the hardness of sponge cake stored in MAP. Another explanation for this result is the recrystallization of sugar after baking. In addition, sensory results showed the acceptability of all cakes except the sample containing 800 ppm CL EOs [38]. The low sensory acceptability of a high dosage of essential oils is another crucial point that limits the direct use of EOs in food products. Therefore, adding these natural antimicrobials to packages seems a suitable method to overcome these disadvantages of EOs [39].

## 4. Conclusions

Maintaining the freshness and good quality of gluten-free bakery and confectionery products is still challenging. In this study, high-in oat beta-glucan, gluten-free yeast-leavened cake was packed in MAP combined with active packaging (various EOs) and stored for 14 days. The evaluation of microbiological and physicochemical properties was done. The biggest issue was decreasing moisture content regardless of the packaging method. The loss of water influenced texture, indicating a significant increase in hardness. The beta-glucan content didn’t change during storage, which is a very positive result. Cake samples were also getting darker, but the saturation of redness and yellowness did not change rapidly. EOs did not influence physicochemical characterization, but the packaging method had a significant impact on those parameters. In the case of texture, the best combination was AE/MAP_1_ and CV/MAP_1_. For color, the best resulting samples were those packed in ATM or MAP_1_ regardless of EOs. Thus, the packaging system was a key element to prolong the shelf life of gluten-free yeast-leavened cake. EOs significantly slowed down the growth of microorganisms regardless of the type of gas mixture. However, the best bacteriostatic effect was in MAP_2_. The best results were obtained for samples packed in MAP_2_ with cinnamon or clove essential oil. Those samples had less than 1.0 log (cfu/g) after 14 days. The most extensive microbial growth was observed in samples with no EOs and in samples packed in ATM. Those results are very promising and show that alternative, natural emitters can be added to inhibit microbial growth. Nevertheless, as the next step, sensory analyses should be performed to confirm their whole usability and to check how the volatile compounds interfere with food.

## Figures and Tables

**Table 1 foods-11-00872-t001:** The total plate count (log (cfu/g)) of GF cake rich (mean ± SD; *n* = 3).

Type of EOs	Type of Gas Mixture	TPC at Storage Day	
		D7	D14
NO	ATM	4.72 ± 0.21 ^iA^	6.24 ± 0.17 ^hB^
	MAP_1_	4.30 ± 0.18 ^hA^	5.73 ± 025 ^gB^
	MAP_2_	2.19 ± 0.14 ^eA^	5.12 ± 0.33 ^fB^
AE	ATM	2.58 ± 0.27 ^fA^	4.55 ± 0.25 ^eB^
	MAP_1_	1.68 ± 0.18 ^dA^	3.01 ± 0.16 ^dB^
	MAP_2_	0.78 ± 0.12 ^bA^	2.48 ± 0.24 ^cB^
CV	ATM	3.12 ± 0.21 ^gA^	5.56 ± 0.31 ^fgB^
	MAP_1_	1.51 ± 0.16 ^dA^	2.21 ± 0.17 ^bB^
	MAP_2_	0.83 ± 0.11 ^bA^	0.94 ± 0.12 ^aA^
CN	ATM	2.32 ± 0.15 ^efA^	3.23 ± 0.19 ^dB^
	MAP_1_	1.28 ± 0.22 ^cA^	2.18 ± 0.21 ^bB^
	MAP_2_	0.54 ± 0.06 ^aA^	0.84 ± 0.14 ^aB^

Letters (a, b…) show the significant differences in a column (*p* ≤ 0.05) and letters (A, B…) show the significant differences in a row (*p* ≤ 0.05). NO—no added essential oil; AE—anise star oil; CV—clove oil; CN—cinnamon bark oil. ATM—approx. 78% N_2_/21% O_2_/0.04% CO_2_; MAP_1_—0% CO_2_/100%/N_2_; MAP_2_—60% CO_2_/40%/N_2_.

**Table 2 foods-11-00872-t002:** The water activity, moisture content [%], and content of beta-glucan [g/100 g] of GF cake (mean ± SD; *n* = 3).

		Water Activity			Moisture Content [%]			Beta-Glucan [g/100 g]		
		D0	D7	D14	D0	D7	D14	D0	D7	D14
non-packed		0.95585 ± 0.06 ^B^*^,C^	-	-	35.04 ± 0.22 ^C^	-	-	1.11 ± 0.04 ^A^	-	-
NO	ATM	-	0.9324 ± 0.0006 ^aA^*	0.934 ± 0.0005 ^bA^*	-	30.52 ± 0.19 ^aB^	26.29 ± 0.24 ^bA^	-	1.15 ± 0.07 ^aA^	1.14 ± 0.11 ^aA^
	MAP_1_	-	0.939 ± 0.0004 ^aB^	0.9236 ± 0.0017 ^aA^	-	31.81 ± 0.16 ^bB^	24.84 ± 0.11 ^aA^	-	1.21 ± 0.12 ^aA^	1.18 ± 0.04 ^aA^
	MAP_2_	-	0.9482 ± 0.0014 ^bB^	0.9373 ± 0.0014 ^bA^	-	33.75 ± 0.32 ^cB^	30.63 ± 0.11 ^dA^	-	1.19 ± 0.05 ^aA^	1.15 ± 0.02 ^aA^
AE	ATM	-	0.9479 ± 0.0021 ^bB^	0.9301 ± 0.0012 ^bA^	-	33.16 ± 0.25 ^cB^	26.67 ± 0.21 ^bA^	-	1.12 ± 0.11 ^aA^	1.11 ± 0.06 ^aA^
	MAP_1_	-	0.9435 ± 0.0012 ^bB^	0.9233 ± 0.0002 ^aA^	-	30.51 ± 0.24 ^aB^	27.39 ± 0.11 ^bA^	-	1.17 ± 0.02 ^aA^	1.11 ± 0.08 ^aA^
	MAP_2_	-	0.9424 ± 0.0009 ^bB^	0.9372 ± 0.011 ^bA^	-	32.15 ± 0.54 ^bB^	29.12 ± 0.96 ^cA^	-	1.14 ± 0.05 ^aA^	1.09 ± 0.10 ^aA^
CV	ATM	-	0.9434 ± 0.001 ^bB^	0.9319 ± 0.001 ^bA^	-	33.07 ± 0.35 ^cB^	28.42 ± 0.29 ^cA^	-	1.10 ± 0.01 ^aA^	1.09 ± 0.11 ^aA^
	MAP_1_	-	0.9479 ± 0.0016 ^bA^*	0.9432 ± 0.0004 ^cA^*	-	31.31 ± 0.11 ^bB^	28.17 ± 0.14 ^cA^	-	1.11 ± 0.08 ^aA^	1.07 ± 0.07 ^aA^
	MAP_2_	-	0.9452 ± 0.0003 ^bA^*	0.9408 ± 0.0015 ^cA^*	-	31.55 ± 0.21 ^bA^	31.24 ± 0.08 ^dA^	-	1.22 ± 0.16 ^aA^	1.15 ± 0.09 ^aA^
CN	ATM	-	0.9448 ± 0.0019 ^bB^	0.9345 ± 0.0012 ^bA^	-	33.03 ± 0.08 ^cB^	28.11 ± 0.14 ^cA^	-	1.15 ± 0.04 ^aA^	1.12 ± 0.07 ^aA^
	MAP_1_	-	0.9472 ± 0.0011 ^bB^	0.9345 ± 0.0009 ^bA^	-	33.02 ± 0.17 ^cB^	29.12 ± 0.21 ^cA^	-	1.13 ± 0.03 ^aA^	1.11 ± 0.19 ^aA^
	MAP_2_	-	0.9444 ± 0.0004 ^bB^	0.9389 ± 0.0008 ^bA^	-	31.42 ± 0.19 ^bB^	30.55 ± 0.19 ^dA^	-	1.18 ± 0.16 ^aA^	1.16 ± 0.09 ^aA^

Letters (a, b…) show the significant differences in a column (*p* ≤ 0.05) and letters (A, B…) show the significant differences in a row (*p* ≤ 0.05). NO—no added essential oil; AE—anise star oil; CV—clove oil; CN—cinnamon bark oil. ATM—approx. 78% N_2_/21% O_2_/0.04% CO_2_; MAP_1_—0% CO_2_/100%/N_2_; MAP_2_—60% CO_2_/40%/N_2_.

**Table 3 foods-11-00872-t003:** Changes in color components L*, a*, and b* during 14-days storage of GF cake (mean ± SD; *n* = 10).

		L*			a*			b*		
		D0	D7	D14	D0	D7	D14	D0	D7	D14
non-packed		64.13 ± 1.68 ^B^	-	-	−1.178 ± 0.23 ^A^	-	-	23.189 ± 0.99 ^A^	-	-
NO	ATM	-	56.62 ± 1.79 ^aA^	56.72 ± 1.91 ^bcdA^	-	−0.60 ± 0.07 ^cdB^	−1.11 ± 0.19 ^aA^	-	24.87 ± 0.8 ^fA^	23.54 ± 0.83 ^cdA^
	MAP_1_	-	62.83 ± 1.21 ^fB^	51.92 ± 1.67 ^aA^	-	−0.63 ± 0.06 ^dB^	−0.32 ± 0.05 ^dC^	-	23.96 ± 0.69 ^deA^	23.47 ± 1.14 ^cdA^
	MAP_2_	-	62.81 ± 0.98 ^fB^	55.01 ± 1.04 ^bA^	-	−0.83 ± 0.02 ^bB^	−0.77 ± 0.11 ^bcB^	-	23.28 ± 1.09 ^bcdA^	23.55 ± 1.4 ^cdA^
AE	ATM	-	65.52 ± 1.12 ^gB^	56.16 ± 1.81 ^bcdA^	-	−0.95 ± 0.12 ^abA^	−0.75 ± 0.03 ^bB^	-	24.61 ± 1.14 ^efA^	24.29 ± 1.04 ^dA^
	MAP_1_	-	59.98 ± 1.05 ^cdA^	57.86 ± 1.07 ^cdA^	-	−0.72 ± 0.02 ^cB^	−1.11 ± 0.11 ^aA^	-	22.88 ± 0.89 ^abcA^	22.45 ± 0.52 ^aA^
	MAP_2_	-	56.91 ± 1.55 ^aA^	55.82 ± 1.83 ^bcA^	-	−0.62 ± 0.04 ^cdB^	−0.74 ± 0.09 ^bcB^	-	22.33 ± 1.58 ^aA^	23.33 ± 1.25 ^abcA^
CV	ATM	-	60.27 ± 1.45 ^cdC^	56.36 ± 1.36 ^bcdB^	-	−0.81 ± 0.08 ^bB^	−1.08 ± 0.23 ^bA^	-	23.11 ± 0.99 ^cdA^	23.06 ± 1.14 ^abA^
	MAP_1_	-	60.74 ± 1.56 ^deB^	57.21 ± 1.19 ^bcdA^	-	−0.42 ± 0.08 ^eB^	−1.05 ± 0.06 ^aA^	-	22.98 ± 0.86 ^abcA^	23.52 ± 1.04 ^cdA^
	MAP_2_	-	57.71 ± 1.37 ^abA^	58.25 ± 1.63 ^dA^	-	−0.63 ± 0.03 ^dB^	−0.61 ± 0.04 ^cB^	-	23.45 ± 1.08 ^cdA^	23.5 ± 0.91 ^cdA^
CN	ATM	-	62.37 ± 1.67 ^fC^	56.71 ± 1.05 ^bcdaB^	-	−1.1 ± 0.20 ^aA^	−0.79 ± 0.07 ^bcB^	-	22.78 ± 0.6 ^abcA^	23.94 ± 0.97 ^cdA^
	MAP_1_	-	63.06 ± 1.30 ^fB^	57.44 ± 1.67 ^cdaA^	-	−0.41 ± 0.06 ^cdC^	−0.71 ± 0.04 ^bB^	-	23.95 ± 0.8 ^deA^	24.19 ± 0.84 ^dA^
	MAP_2_	-	57.75 ± 1.11 ^bAc^	55.01 ± 1.56 ^baA^	-	−0.59 ± 0.06 ^cdB^	−0.03 ± 0.01 ^eC^	-	22.42 ± 0.76 ^abA^	24.2 ± 1.55 ^dA^

Letters (a, b…) show the significant differences in a column (*p* ≤ 0.05) and letters (A, B…) show the significant differences in a row (*p* ≤ 0.05). NO—no added essential oil; AE—anise star oil; CV—clove oil; CN—cinnamon bark oil. ATM—approx. 78% N_2_/21% O_2_/0.04% CO_2_; MAP_1_—0% CO_2_/100%/N_2_; MAP_2_—60% CO_2_/40%/N_2_.

**Table 4 foods-11-00872-t004:** Changes in hardness and springiness during 14-days storage of GF cake (mean ± SD; *n* = 10).

		Hardness [N]			Springiness [-]		
		D0	D7	D14	D0	D7	D14
non-packed		7.77 ± 0.90 ^A^	-	-	0.57 ± 0.01 ^B^*^,C^	-	-
NO	ATM	-	19.75 ± 1.99 ^cB^	28.01 ± 1.56 ^cC^	-	0.18 ± 0.04 ^abA^*	0.63 ± 0.04 ^dC^*
	MAP_1_	-	19.14 ± 4.4 ^cdB^	27.49 ± 3.02 ^cC^	-	0.47 ± 0.09 ^cdB^	0.32 ± 0.01 ^abA^
	MAP_2_	-	16.72 ± 1.99 ^bcdB^	25.92 ± 2.78 ^bcC^	-	0.43 ± 0.09 ^cdB^	0.30 ± 0.03 ^aA^
AE	ATM	-	15.65 ± 1.28 ^bB^	29.03 ± 2.19 ^cC^	-	0.49 ± 0.05 ^cdB^	0.33 ± 0.09 ^abA^
	MAP_1_	-	18.23 ± 1.2 ^cB^	18.12 ± 3.55 ^aB^	-	0.19 ± 0.07 ^abA^	0.36 ± 0.05 ^bB^
	MAP_2_	-	12.49 ± 1.19 ^aB^	20.37 ± 2.78 ^aC^	-	0.16 ± 0.07 ^aA^	0.41 ± 0.05 ^bcB^
CV	ATM	-	18.27 ± 2.01 ^cB^	25.91 ± 2.11 ^bcC^	-	0.37 ± 0.06 ^bcA^*	0.52 ± 0.08 ^cB^*
	MAP_1_	-	16.25 ± 2.28 ^bcB^	17.24 ± 2.45 ^aB^	-	0.58 ± 0.02 ^dB^*	0.33 ± 0.08 ^abA^*
	MAP_2_	-	16.92 ± 2.77 ^bcB^	26.42 ± 3.64 ^cC^	-	0.24 ± 0.07 ^bA^	0.44 ± 0.07 ^bcBC^
CN	ATM	-	19.23 ± 2.83 ^cdB^	36.61 ± 3.56 ^dC^	-	0.47 ± 0.07 ^cB^	0.29 ± 0.09 ^aA^
	MAP_1_	-	23.23 ± 2.02 ^dB^	21.02 ± 3.34 ^abB^	-	0.55 ± 0.03 ^dB^*	0.39 ± 0.02 ^abA^*
	MAP_2_	-	22.27 ± 1.49 ^dB^	19.20 ± 2.04 ^aB^	-	0.33 ± 0.09 ^bcA^*	0.30 ± 0.09 ^aA^*

Letters (a, b…) show the significant differences in a column (*p* ≤ 0.05) and letters (A, B…) show the significant differences in a row (*p* ≤ 0.05). NO—no added essential oil; AE—anise star oil; CV—clove oil; CN—cinnamon bark oil. ATM—approx. 78% N_2_/21% O_2_/0.04% CO_2_; MAP_1_—0% CO_2_/100%/N_2_; MAP_2_—60% CO_2_/40%/N_2_.

## Data Availability

The datasets generated for this study are available on request to the corresponding author.

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
