# Peer review of "Evaluation of Modified Atmosphere Packaging in Combination with Active Packaging to Increase Shelf Life of High-in Beta-Glucan Gluten Free Cake"

_foods, 2022, doi:10.3390/foods11060872_

Round 1

Reviewer 1 Report

Aim and novelty:

The active packaging incorporated with EO was used for prolonging the shelf life of the food product and EO is also known for the inhibitory activity against different spoilage pathogens, Then, what is the rationale behind using the additional treatment MAP (modified atmosphere condition) to prolong the shelf life of the food product.

  • Authors need to justify and proved by antimicrobial results how many days shelf-life of GF cake was increased with the combination of active, and MAP packaging compared to only active packaging (with EO).

Methodology: The methodology section is very poorly written and not clearly described.

  • Such as: In Packaging: how authors have done the packaging of baked GF cake into PET/CPP+HF film? Have authors developed films with essential oils? What percentage of essential oils was incorporated into the film? All this information is missing.
  • Authors need to justify if an essential oil is not incorporated into the film, as there is a high chance of migrating oils into the GF cake and there is less likely to get buyers of essential oil flavored GF cakes…
  • Microbiological test: Authors need to provide a brief description of the testing method “Test was done according to PN-EN ISO 4833-1:2013-12”.
  • How have authors done the moisture content and water activity test, Beta-glucan content analysis, Texture profile analysis?
  • In the Color analysis, what is L*a*b*? explain them..

Result:

Results are presented in very confusing way and hard to follow the results in table 2 “The ranges of total number of microbials were divided as follow: 0 – ≤ 1.0 log (cfu/g), X – > 1.0 log – ≤ 3.0 log (cfu/g), XX – >3.0 log – ≤ 5.0 170 log (cfu/g), XXX – > 5.0 log (cfu/g)”

Authors should be presented the microbiological results in Log value, and shows the log reduction with respect to the control sample?

Author Response

Thank you very much for your insightful comments on our paper again. The following parts are our replies to all your questions after deliberation.

Comment 1: Aim and novelty: The active packaging incorporated with EO was used for prolonging the shelf life of the food product and EO is also known for the inhibitory activity against different spoilage pathogens, Then, what is the rationale behind using the additional treatment MAP (modified atmosphere condition) to prolong the shelf life of the food product.

Reply: The combination of EOs with modified atmosphere packaging was aimed at testing the synergistic effect of these factors, because, as many publications indicate (e.g. Matam, et al. 2006; Gutiérrez et al. 2011), the individual influence of these factors is not satisfactory to inhibit the growth of microorganisms.

Comment 2: Authors need to justify and proved by antimicrobial results how many days shelf-life of GF cake was increased with the combination of active, and MAP packaging compared to only active packaging (with EO).

Reply: Changed as suggested by the reviewer.

Comment 3: Methodology: The methodology section is very poorly written and not clearly described. Such as: In Packaging: how authors have done the packaging of baked GF cake into PET/CPP+HF film? Have authors developed films with essential oils? What percentage of essential oils was incorporated into the film? All this information is missing.

Reply: No, we have not developed films with essential oils. We used PET/PE trays with attached rectangle of filter papers with dropps (500ul) essential oils (Section 2.3)

Comment 4: Authors need to justify if an essential oil is not incorporated into the film, as there is a high chance of migrating oils into the GF cake and there is less likely to get buyers of essential oil flavored GF cakes…

Reply: Yes, of course, the aroma of the oil was felt direct after opening the package, but in our opinion, the amount of oil that was used did not disqualify the product because about 10 min after opening package the aroma intensity strongly decreases.  It is important to emphasize  that most people have positive associations with the smell of these spices. Moreover, it should be noted that sensory analysis was not performed at this stage of the study, as the primary focus was on demonstrating the positive effect of using the essential oil (i.e. cloves, anise and cinnamon) as active component in packaging on extending the shelf life of gluten-free cake. We planing a semi-consumer assessment to next article.

Comment 5: Microbiological test: Authors need to provide a brief description of the testing method “Test was done according to PN-EN ISO 4833-1:2013-12”.

Reply: Changed as suggested by the reviewer (section 2.4).

Comment 6: How have authors done the moisture content and water activity test, Beta-glucan content analysis, Texture profile analysis?

Reply: Changed as suggested by the reviewer. The methodologies have been detailed.

Comment 7: In the Color analysis, what is L*a*b*? explain them..

Reply: Changed as suggested by the reviewer.

Comment 8: Result: Results are presented in very confusing way and hard to follow the results in table 2 “The ranges of total number of microbials were divided as follow: 0 – ≤ 1.0 log (cfu/g), X – > 1.0 log – ≤ 3.0 log (cfu/g), XX – >3.0 log – ≤ 5.0 170 log (cfu/g), XXX – > 5.0 log (cfu/g)”. Authors should be presented the microbiological results in Log value, and shows the log reduction with respect to the control sample?

Reply: Changed as suggested by the reviewer.

Reviewer 2 Report

The application of EOs based active packaging combined with MAP is not new, even though, as reported by the authors, most of them concern meat products. This scenario makes this paper very interesting for the field. However, authors should deeply improve the quality of the manuscript to make it publishable for this journal. First of all the quality of the language is very poor so the authors should submit the paper for English revision. Moreover, microbiological aspects are totally neglected. It’s not acceptable that results of microbial contamination are given as intervals of counting (2 Log cycles), authors must report the result of plate counting and the number of replications should be given. Indeed, it seems results of table 3 and 4 are averages, but of how many replicates? is the standard deviation or standard error after the average? It’s totally unclear the results of statistical analysis in the tables.

Lines 140-141: authors write “The microbiology analysis was carried out to examine the ability of used EOs to inhibit the growth of pathogens ….” but there is no evidence in the experiment. Authors could carry out a challenge test.

Author Response

Thank you very much for your insightful comments on our paper again. The following parts are our replies to all your questions after deliberation.

Comment 1: The application of EOs based active packaging combined with MAP is not new, even though, as reported by the authors, most of them concern meat products. This scenario makes this paper very interesting for the field. However, authors should deeply improve the quality of the manuscript to make it publishable for this journal. First of all the quality of the language is very poor so the authors should submit the paper for English revision.

Reply: The current version of the manuscript is English proofreading.

Comment 2: Moreover, microbiological aspects are totally neglected. It’s not acceptable that results of microbial contamination are given as intervals of counting (2 Log cycles), authors must report the result of plate counting and the number of replications should be given.

Reply: Changed as suggested by the reviewer.

Comment 3: Indeed, it seems results of table 3 and 4 are averages, but of how many replicates? is the standard deviation or standard error after the average? It’s totally unclear the results of statistical analysis in the tables.

Reply: Changed as suggested by the reviewer. Tables have been reorganized.

Comment 4: Lines 140-141: authors write “The microbiology analysis was carried out to examine the ability of used EOs to inhibit the growth of pathogens ….” but there is no evidence in the experiment. Authors could carry out a challenge test.

Reply: Changed as suggested by the reviewer.

Reviewer 3 Report

This manuscript provides finding on shelf-life extension of gluten free cake. There are some points that need to be reconsidered.

All the subscript and superscript should be revised e.g. L16, L123, L125

Abstract

L17 should be 0.04 CO2

L19 There should be more key findings in the abstract or more detail of the measurement e.g. L21 Physicochemical quality consisted of …?

Introduction

L71 Any reason that GF cake will give diverse results to those typical cake?

Materials and methods

L122 What is HF?

L126 should be 0.04 CO2

L127 This means gas mixing prior to flushing? Please give more detail about the gas flushing system.

Table 1 The left column shows repeat values. Are they necessary to show this column? The gas ratio should be stated in this Table.

L139 Any sample preparation before the measurement in section 2.4-2.6? What about the storage condition? How many replicates?

L155 Please add detail of the measurement e.g. probe size, speed.

Results and discussion

L174 CO2 had bacteriostatic activity, while aerobic microorganisms and mold are also sensitive to high CO2 atmosphere (Kimbuathong 2020 Food Chemistry).

L185 Release of essential oils to the package headspace caused absorption of the essential oils on food surface which delayed growth of microorganisms (Laorenza 2021 Food Chemistry). Moreover, the thin layer of absorbed essential oils covering food surface possibly interacted and reduced oxygen diffusion and delayed the growth of microorganisms (Klinmalai 2021 LWT).

L200 Remove “In study”

Table 2 Why don’t the authors show number? The microbial count should be stated as number (mean ± SD).

L212 What is “the nearest possible water activity”?

L220 MAP 2 which consisted of high CO2 tended to retain moisture in the cake which was coincident with the lowest microbial growth.

Table 3 and 4 “…difference in a row..”? It should be specific as L* cannot be compared with a*. It should be difference between storage time.

L282 Packaging and storage conditions significantly affected hardness of bakery and starch based products. Difference in relative vapor pressure between food and the headspace is the major driving force of moisture transfer and loss of moisture which induced increasing hardness (Wangprasertkul 2021 Food Control; Bumbudsanpharoke 2022 Food Control).

Author Response

Thank you very much for your insightful comments on our paper again. The following parts are our replies to all your questions after deliberation.

Comment 1: All the subscript and superscript should be revised e.g. L16, L123, L125

Reply: Changed as suggested by the reviewer.

Comment 2: Abstract: L17 should be 0.04 CO2

Reply: Changed as suggested by the reviewer.

Comment 2: L19 There should be more key findings in the abstract or more detail of the measurement e.g.

Reply: Changed as suggested by the reviewer.

Comment 3: L21 Physicochemical quality consisted of …?

Reply: It has been verified.

Comment 4: Introduction: L71 Any reason that GF cake will give diverse results to those typical cake?

Reply: We have not researched this in this article. This article is a continuation of previous research in which a clean label GF cake recipe was developed. Generally, the GF cake can have a higher moisture content and Aw value because various hydrocolloids are used which have high water absorption and thus be a better medium for microbes. We used oat high-fiber powder. This is explained in section 3.1.

Comment 5: Materials and methods: L122 What is HF?

Reply: There was a mistake, there should be AF (AF laminate). It was corrected in section 2.3.

Comment 6: L126 should be 0.04 CO2

Reply: Changed as suggested by the reviewer.

Comment 7: L127 This means gas mixing prior to flushing? Please give more detail about the gas flushing system.

Reply: The individual gases (CO2 and N2) were combined and mixed in defined proportions (for MAP1 or MAP2) in a gas mixer to obtain a gas mixture which was then used to fill the package with the product inside.

Comment 8: Table 1 The left column shows repeat values. Are they necessary to show this column? The gas ratio should be stated in this Table.

Reply: All tables have been reorganized and the previous table 1 has been excluded.

Comment 9: L139 Any sample preparation before the measurement in section 2.4-2.6? What about the storage condition? How many replicates?

Reply: Changed as suggested by the reviewer.

Comment 10: L155 Please add detail of the measurement e.g. probe size, speed.

Reply: Changed as suggested by the reviewer.

Comment 11: Results and discussion: L174 CO2 had bacteriostatic activity, while aerobic microorganisms and mold are also sensitive to high CO2 atmosphere (Kimbuathong 2020 Food Chemistry).

Reply: Changed as suggested by the reviewer.

Comment 12: L185 Release of essential oils to the package headspace caused absorption of the essential oils on food surface which delayed growth of microorganisms (Laorenza 2021 Food Chemistry). Moreover, the thin layer of absorbed essential oils covering food surface possibly interacted and reduced oxygen diffusion and delayed the growth of microorganisms (Klinmalai 2021 LWT).

Reply: Changed as suggested by the reviewer.

Comment 13: L200 Remove “In study”

Reply: Changed as suggested by the reviewer.

Comment 14: Table 2 Why don’t the authors show number? The microbial count should be stated as number (mean ± SD).

Reply: Changed as suggested by the reviewer. Table (currently 1) include mean +-SD of total plate count (log (cfu/g)).

Comment 15: L212 What is “the nearest possible water activity”?

Reply: Changed as suggested by the reviewer.

Comment 16: L220 MAP 2 which consisted of high CO2 tended to retain moisture in the cake which was coincident with the lowest microbial growth.

Reply: Added as suggested by the reviewer: A higher tended to retain moisture during storage in a package with a higher CO2 (MAP2) content was observed, however, no greater microbial growth was observed. The presence of CO2 could have a greater bacteriostatic effect than the lower moisture content. This may be because CO2 dissociates to a greater extent with a higher moisture content in the product, which leads to the formation of carbonic acid on the product surface and a reduce of the pH value.

Comment 17: Table 3 and 4 “…difference in a row..”? It should be specific as L* cannot be compared with a*. It should be difference between storage time.

Reply: All tables have been reorganized as suggested by the reviewers.

Comment 18: L282 Packaging and storage conditions significantly affected hardness of bakery and starch based products. Difference in relative vapor pressure between food and the headspace is the major driving force of moisture transfer and loss of moisture which induced increasing hardness (Wangprasertkul 2021 Food Control; Bumbudsanpharoke 2022 Food Control).

Reply: Changed as suggested by the reviewer.

Reviewer 4 Report

Dear authors

You will find my comments regarding your research project, I hope they will be useful for the improvement of this 

Take care of the subscripts in N2, CO2, O2.

The keywords must be completely different from those used in the title. This to give greater visibility to your paper.

In the introduction, expand the discussion on the combined strategies that have been used for the conservation of cake and why it was decided to work with the modified atmosphere and the active container.

The general objective and the working hypothesis should be mentioned in the introduction.

Materials and methods

In section 2.2 make a table specifying the composition of the cake.

In section 2.3 specify the power of the UV lamp.

In section 2.3 take care of superscripts, these errors are found throughout the text.

In section 2.4, place at least the basis of the technique used.

In section 2.5, how was the equipment calibrated?

All equipment and software used must contain the city and country, example

- Shimadzu UV-1800 spectrometer (Shimadzu Inc., Kyoto, Japan)

Section 2.8 separate the analysis of mechanical properties and the colorimetric analysis. You did not carry out a texture profile analysis, you carried out a unidirectional compression test, where the maximum hardness obtained (hardness) is reported as firmness. Review foundation of the Instron.

Results

Perform the colorimetry analysis based on the total change in color or chromaticity. This gives greater impact to the investigation.

In section 2.8 mention whether statistically significant differences were evaluated based on the combination of treatments, storage time or both. How many replicates were used for the analysis of each test?

In the microbiological analysis, change the nomenclature, if the plate count was performed, if replicates were performed, report the mean together with its standard deviation. This is so that the reader can check the evolution of the cake during storage in the table.

Correlate moisture content and water activity with microbiological analysis.

Correlate color profile parameters with moisture changes in samples.

The conclusions must respond to the general objective of the work and must also reflect the testing of hypotheses.

Author Response

Thank you very much for your insightful comments on our paper again. The following parts are our replies to all your questions after deliberation.

Comment 1: Take care of the subscripts in N2, CO2, O2.

Reply: Changed as suggested by the reviewer.

Comment 2: The keywords must be completely different from those used in the title. This to give greater visibility to your paper.

Reply: Changed as suggested by the reviewer.

Comment 3: In the introduction, expand the discussion on the combined strategies that have been used for the conservation of cake and why it was decided to work with the modified atmosphere and the active container.

Reply: Changed as suggested by the reviewer.

Comment 4: The general objective and the working hypothesis should be mentioned in the introduction.

Reply: Changed as suggested by the reviewer.

Comment 5: Materials and methods: In section 2.2 make a table specifying the composition of the cake.

Reply: I agree with the reviewer that the GF cake composition could be in the table, but for only one cake recipe, the description of the cake composition in section 2.2 is sufficient and clear.

Comment 6: In section 2.3 specify the power of the UV lamp.

Reply: Added as suggested by the reviewer.

Comment 7: In section 2.3 take care of superscripts, these errors are found throughout the text.

Reply: Changed as suggested by the reviewer.

Comment 8: In section 2.4, place at least the basis of the technique used.

Reply: Changed as suggested by the reviewer.

Comment 9: In section 2.5, how was the equipment calibrated?

Reply: Moisture analyzer and water activity meter were calibrated according to manual these aparatures.

Comment 10: All equipment and software used must contain the city and country, example Shimadzu UV-1800 spectrometer (Shimadzu Inc., Kyoto, Japan)

Reply: Changed as suggested by the reviewer.

Comment 12: Section 2.8 separate the analysis of mechanical properties and the colorimetric analysis. You did not carry out a texture profile analysis, you carried out a unidirectional compression test, where the maximum hardness obtained (hardness) is reported as firmness. Review foundation of the Instron.

Reply: The mechanical properties and colorimetric analyzes were divided into 2 sections. From the TPA test results, only 2 selected parameters were analyzed. Changed firmness to hardness throughout the manuscript as suggested by the reviewer.

Comment 13: Results: Perform the colorimetry analysis based on the total change in color or chromaticity. This gives greater impact to the investigation.

Reply: Yes, delta E gives information about the total color difference, however it does not indicate which color coordinate is most responsible for this difference. The table is extensive and in my opinion there is no need to expand it.

Comment 14: In section 2.8 mention whether statistically significant differences were evaluated based on the combination of treatments, storage time or both. How many replicates were used for the analysis of each test?

Reply: Changed as suggested by the reviewer (currently Section 2.9.)

Comment 15: In the microbiological analysis, change the nomenclature, if the plate count was performed, if replicates were performed, report the mean together with its standard deviation. This is so that the reader can check the evolution of the cake during storage in the table.

Reply: Changed as suggested by the reviewer (table currently 1).

Comment 16: Correlate moisture content and water activity with microbiological analysis.

Reply: These corelations are poor (r <0.2), so I did not include them in the article.

Comment 17: Correlate color profile parameters with moisture changes in samples.

Reply: These corelations are poor (r <0.2), so I did not include them in the article.

Comment 18: The conclusions must respond to the general objective of the work and must also reflect the testing of hypotheses.

Reply: Changed as suggested by the reviewer. The conclusions respond to the general objective of the work and also reflect the testing of hypotheses.

Round 2

Reviewer 1 Report

The authors have revised the manuscript significantly and responded to the reviewer's comments clearly in the response sheet.

Author Response

Comment: The authors have revised the manuscript significantly and responded to the reviewer's comments clearly in the response sheet.

Reply: Thank you very much for your insightful comments on our paper again.

Reviewer 2 Report

How the authors found the level of contamination reported in the table 2 after the oven cooking at 160°C for 40 min. It seems too much high. 
They should report the residual bacterial population immediately after the cooking.
Moreover, the author insist to cite the activity against the pathogens, but no evidence is given about this.

Author Response

Thank you very much for your insightful comments on our paper again. The following parts are our replies to all your questions after deliberation.

Comment 1: How the authors found the level of contamination reported in the table 2 after the oven cooking at 160°C for 40 min. It seems too much high. 

Reply: The GF cake was cooled down in the conditions of the technological laboratory surrounding the room and then contamination could occur from the ambient air (we did not use laminar air flow). We did not sterilize the surface of the cake by exposing them to UV light to eliminate surface contamination. We sterilized only the packaging. This procedure provided better conditions for testing the antimicrobiological properties of essential oil.

Comment 2: They should report the residual bacterial population immediately after the cooking.
Moreover, the author insist to cite the activity against the pathogens, but no evidence is given about this.

Reply: We did not perform a microbiological analysis of the sample immediately after heat treatment, because such a product is fresh and there is no microbiological hazard with such a product. Only after storage does a microbiological analysis make sense.

We focused on the analysis of the TPC value as an indicator of microbial growth reduction by the type of packaging used and the type of essential oil.

This manuscript was proofread by a English native speaker.

Reviewer 3 Report

The manuscript has been revised as recommend. Only 1 point to be revised

Ref. [37] should be replaced with this one 

https://doi.org/10.1016/j.foodcont.2021.108541

Author Response

Comment 1: The manuscript has been revised as recommend. Only 1 point to be revised

Ref. [37] should be replaced with this one https://doi.org/10.1016/j.foodcont.2021.108541

Reply: Changed as suggested by the reviewer.

Reviewer 4 Report

Dear authors.

I am grateful that you have made the comments raised in the review.

Author Response

Comment: Dear authors. I am grateful that you have made the comments raised in the review.

Reply: Thank you very much for your insightful comments on our paper again.